# Segregation Behavior of Miscible PC/PMMA Blends during Injection Molding

**DOI:** 10.3390/ma15092994

**Published:** 2022-04-20

**Authors:** Nantina Moonprasith, Jitsuhiro Date, Takumi Sako, Takumitsu Kida, Tatsuhiro Hiraoka, Masayuki Yamaguchi

**Affiliations:** 1School of Materials Science, Japan Advanced Institute of Science and Technology, 1-1 Asahidai, Nomi 923-1292, Japan; moonprasith.n@jaist.ac.jp (N.M.); s1710114@jaist.ac.jp (J.D.); s-takumi@jaist.ac.jp (T.S.); tkida@jaist.ac.jp (T.K.); 2Sirindhorn International Institute of Technology, Thammasat University, 99 Moo 18, Paholyothin, Khlong Luang 12120, Thailand; 3Hiroshima R&D Center, Mitsubishi Chemical Corporation, 20-1, Miyukicho, Otake, Hiroshima 739-0693, Japan; hiraoka.tatsuhiro.ma@m-chemical.co.jp

**Keywords:** polymer blends, viscoelastic properties, injection molding, segregation, polycarbonate, poly(methyl methacrylate)

## Abstract

Miscible blends composed of bisphenol-A polycarbonate (PC) and poly(methyl methacrylate) (PMMA), in which one of them has low molecular weight, were employed to study the surface segregation behavior during flow. The blend samples showed typical rheological behaviors, such as simple polymer melts without a long-time relaxation mechanism ascribed to phase separation, demonstrating that they were miscible. After injection molding, the amounts of a low molecular weight component on the blend surface were found to be larger than the actual blend ratio. Because the injection-molded products were transparent despite a huge difference in refractive indices between PC and PMMA, they showed no phase separation. This result demonstrated that surface segregation of a low molecular weight component occurred under flow field, which expands the material design such as tough plastics with good scratch resistance and optical fibers with tapered refractive index.

## 1. Introduction

Bisphenol-A polycarbonate (PC) and poly(methyl methacrylate) (PMMA) are widely used as plastic glasses because they are both highly transparent. However, their properties are significantly different. For example, PC has a low surface hardness and poor scratch resistance, excellent mechanical toughness, good heat resistance, poor weatherability, and a high refractive index with large birefringence. In contrast, PMMA has a high surface hardness and excellent scratch resistance, poor toughness, poor heat resistance, good weatherability, and a low refractive index with little birefringence. Therefore, it is desirable to compensate for the defects of one polymer by adding a different one. In general, however, blends of PC and PMMA exhibit phase separation and consequent loss of transparency due to the huge difference in the refractive indices of the components (PC 1.57 and PMMA 1.49) [1,2,3,4,5,6,7]. Although this makes it easier to study the miscibility of PC/PMMA blends, the industrial application is greatly restricted. When either component has low molecular weight, however, they are miscible [2,3,5,8,9,10]. This is attributed to a low positive value of Flory–Huggins interaction parameter [6]. The blend system has a phase diagram with a lower critical solution temperature (LCST). Furthermore, segregation behavior, i.e., a concentration gradient, has been detected without phase separation in miscible blends comprising PC and low molecular weight PMMA under a velocity gradient [8] and under a temperature gradient [9]. Although there have been numerous papers on the flow-induced phase mixing and/or demixing of polymer blends [10,11,12,13,14,15], to the best of our knowledge, only a few researchers have reported the segregation of miscible blends under flow field [8,9,16,17]. The surface segregation behavior was also detected on blend samples exposed to air atmosphere, in which surface free energy plays an important role [18,19,20,21,22]. Moreover, the segregation occurred in the amorphous region of miscible blends during crystallization of one component [23,24,25], which must be different from the flow-induced segregation.

PC/PMMA blends with a concentration gradient are attractive because PC and PMMA have significantly different properties, as mentioned. Therefore, various material designs can be proposed without losing transparency. In particular, the effect of flow fields on segregation behavior is quite important because they are readily created in conventional processing machines. For example, a tough injection-molded product with good scratch resistance can be obtained from a PC/PMMA blend when the core is rich in PC and the surface is rich in PMMA [26,27]. An optical fiber with a refractive index gradient produced by extrusion is another example because it can reduce light loss [28,29].

In the present study, the surface segregation behavior of injection-molded products comprising PC and PMMA, in which one component is of low molecular weight, was studied.

## 2. Materials and Methods

### 2.1. Materials

Two types of PC and two types of PMMA with different molecular weights were used. H and L in the sample codes represent high and low molecular weights, respectively. For example, PMMA-H represents the poly(methyl methacrylate) sample with the higher molecular weight. PC-H, PC-L, and PMMA-H are commercially available polymers, and PMMA-L was prepared by Mitsubishi Chemical Corporation in Tokyo, Japan. The number-average (*M_n_*) and weight-average (*M_w_*) molecular weights, evaluated by size exclusion chromatography (HLC-8020; Tosoh, Tokyo, Japan) using a polystyrene standard, are summarized in Table 1. The viscoelastic properties of PC-H and PMMA-H have been described elsewhere [8,9,30].

### 2.2. Sample Preparation

Prior to melt blending, the polymers were dried at 80 °C for 4 h under vacuum. The samples were melt-mixed using a co-rotating twin-screw extruder (ULT15TWnano; Technovel, Osaka, Japan). The temperature was maintained at 250 °C, and the screw rotation speed was 30 rpm. The blend ratios of the samples were 90/10 and 80/20 (wt/wt), in which the low molecular weight components constituted 10 and 20 wt.%. For comparison, PC-H/PMMA-H (90/10) and PMMA-H/PC-H (90/10) were also prepared by the same procedure.

Compression molding was performed at 250 °C and 10 MPa. This was followed by quenching at 25 °C to obtain 500-µm-thick films. Injection molding was carried out using an injection molding machine (HM7; Nissei Plastic Industrial, Hanishina, Japan). The nozzle/barrel and mold temperatures were maintained at 250 and 70 °C, respectively. Injection-molded bars with the following dimensions were produced: length, 70 mm; width, 10 mm; and thickness, 2 mm. The molten polymer was injected from a square gate measuring 1.5 mm × 1.5 mm.

### 2.3. Measurements

The transparency was evaluated using a UV-vis spectrophotometer (Lamda25; Perkin-Elmer, Waltham, MA, USA) at 589 nm. Compression-molded films (500-mm-thick) were used.

The temperature dependence of the oscillatory tensile modulus at 10 Hz was determined by raising the temperature from 30 to 180 °C at a rate of 2 °C/min using a dynamic mechanical analyzer (E-4000; UBM, Muko, Japan).

The frequency dependence of the oscillatory shear modulus at 250 °C was measured in a molten state using a cone-and-plate rheometer (AR2000ex; TA Instrument, New Castle, DE, USA) under a nitrogen atmosphere. The cone diameter was 25 mm and the cone angle was 4°.

The blend compositions at the surfaces of injection-molded bars were determined by obtaining attenuated total reflectance Fourier-transform infrared (ATR-IR) spectra (Spectrum 100 FT-IR spectrometer; Perkin-Elmer) at 25 °C using KRS-5 as an ATR crystal. The blend samples of various compositions, prepared by compression molding, were also investigated to obtain a calibration curve to evaluate the PMMA content [9]. Figure 1 illustrates the measurement points on an injection-molded bar with the gate position indicated.

## 3. Results and Discussion

### 3.1. Viscoelastic Properties and Miscibility

Miscibility between PC and PMMA has been evaluated by light scattering [1,2,3,4,5,6,7,8,9,10]. There is a possibility to show good transparency of phase-separated blends when the component polymers show the same refractive indices [15,31]. However, the huge difference in the refractive index between PC and PMMA always resulted in light scattering when phase separation occurs. Therefore, it is highly difficult to obtain a transparent blend using only conventional polymers with high molecular weight. Figure 2 shows the 500-µm-thick films prepared by compression molding. The films containing PC-L or PMMA-L were transparent, suggesting that they were miscible, at least at the compression-molding temperature, i.e., 250 °C. According to our recent study using the PC/PMMA blends with similar molecular weights to the present PC-H/PMMA-L, the LCST of the blend containing 20 wt.% of PMMA was 270 °C [10].

The light transmittance values of the films were determined using a UV spectrophotometer, and are included in the figure. The values of the films containing PC-L or PMMA-L (86–89%) were almost the same as those of pure PC-H (86%) and PMMA-H (89%). The small difference between the PC-H blends and the PMMA-H blends must be attributed to the surface reflectance, which can be calculated by Equation (1) [32]. Because approximately 10% of the light was reflected at both surfaces [31], almost no light scattering occurred in the films. In contrast, PC-H/PMMA-H (90/10) and PMMA-H/PC-H (90/10) were opaque owing to light scattering by the phase-separated structure.
(1)$$R_{0}={\left(\frac{n_{film}-n_{air}}{n_{film}+n_{air}}\right)}^{2}$$
where *R*_0_ is the reflectivity and *n_film_* and *n_air_* are the refractive indices of a film and air (≈1).

Miscibility between PC and PMMA has been also studied by dynamic mechanical analysis, because the glass-to-rubber transition is clearly detected for the system. Figure 3 shows the temperature dependencies of the tensile storage modulus *E′* and the loss modulus *E″* of PC-H, PC-H/PMMA-L (90/10), and PC-H/PMMA-L (80/20). The samples were prepared by compression molding. Glass-to-rubber transition was obvious in all the samples. A single peak was detected in the *E″* curve, suggesting that the systems were miscible. It corresponded with Figure 2. The peak temperature, i.e., the glass transition temperature *T_g_*, decreased with increasing PMMA-L content; 163 °C for PC-H, 155 °C for PC-H/PMMA-L (90/10), and 145 °C for PC-H/PMMA-L (80/20). Therefore, it can be concluded that the system is fully miscible, which agrees with previous studies using PC/PMMA blends in which molecular weight of one component was low [2,8,9,10]. It is well known that *T_g_* of a miscible blend follows the Fox equation [33], shown in Equation (2):(2)$$\frac{1}{T_{g-blend}}=\frac{w_{A}}{T_{g-A}}+\frac{w_{B}}{T_{g-B}}$$
where *T_g_*_-*i*_ and *w_i_
*are the *T_g_* and weight fraction of *i*-th component, respectively.

Although *T_g_* of PMMA-L was not detected by the measurement due to the brittle fracture of the film, Equation (2) predicted that it was 86 °C. The *T_g_* value was lower than that of a conventional PMMA [30,31], which is attributed to the low molecular weight as described in detail later. In the case of PC-H/PMMA-H (90/10), i.e., the opaque blend, double peaks were detected in the *E”* curve (but not presented here).

Furthermore, the *E**′* values in the glassy region increased following the addition of PMMA-L, indicating that PMMA-L acted as an anti-plasticizer for PC-H. Various materials are known to act as anti-plasticizers for conventional PC samples; they reduce thermal expansion and modify the birefringence [30,34]. Modulus enhancement would also be desirable for PC [30,35,36,37]. The *E″* values in the glassy region increased with PMMA-L content. This can be attributed to the *b*-dispersion of PMMA [37,38,39,40].

Figure 4 shows the dynamic tensile moduli of PMMA-H, PMMA-H/PC-L (90/10), and PMMA-H/PC-L (80/20). The blends produced a single peak in the *E″* curve. The peak temperatures were lower than that of pure PMMA-H. This can be attributed to low *T_g_* of PC-L. As demonstrated by the Fox–Ferry equation [41], *T_g_* is a function of molecular weight. Because the present sample, PC-L, had a very low molecular weight, *T_g_* was much lower than that of conventional PC samples. Furthermore, the *E**′* values in the glassy region decreased with PC-L content, in contrast to the anti-plasticized system, i.e., PC-H/PMMA-L.

The miscibility of the blend was also investigated by determining its viscoelastic properties in the molten state. Figure 5 shows the dependence on the angular frequency *w* of the oscillatory shear moduli, i.e., the storage modulus *G′* and the loss modulus *G″*. A rheological terminal zone was detected in all the samples. It should be noted that the *G′* values of the blends were lower than those of PC-H and PMMA-H. Furthermore, the gradients of the slopes were close to 2. Therefore, it can be concluded that the blend samples showed typical rheological behaviors as simple polymer melts without a long-time relaxation mechanism ascribed to phase separation, demonstrating that they were miscible. The corresponding Han’s plot [42], i.e., *G′* plotted against *G″*, is shown in Figure 6. The straight lines were confirmed in the plot for both blends with a slightly low position as compared with the lines of the pure materials. These results also suggest that they were miscible at 250 °C.

The zero-shear viscosities *η*_0_ at 250 °C were determined from Equation (3), because the slopes of the *G″* curves were 1 for all samples.
(3)$$\eta_{0}=\underset{\omega \to 0}{\lim}\frac{G^{\prime \prime }}{\omega }$$

The values were 3000 (Pa s) for PC-H and 1950 (Pa s) for PC-H/PMMA-L (90/10). Therefore, the addition of 10 wt.% PMMA-L decreased the *η*_0_ by approximately 35%. The *η*_0_ of case of PMMA-H was 3100 (Pa s), whereas that of PMMA-H/PC-L (90/10) was 1950 (Pa s), i.e., a 37% decrease.

The *η*_0_ values of PMMA-L and PC-L were 3.4 (Pa s) and 2.5 (Pa s), respectively, although the results are not shown in the figures.

Because the systems were miscible, PC-L and PMMA-L acted as plasticizers at 250 °C. According to the Berry–Fox formula [43], presented in Equation (4), *η*_0_ is determined by the volume fraction of the polymer, i.e., PMMA-H or PC-H.
(4)$$\eta_{0}\left(\phi \right)\propto \zeta_{0}\phi^{3.6}$$
where *z*_0_ is the monomeric frictional coefficient and *f* is the volume fraction of the polymer.

Assuming that the melt density of PMMA is the same as that of PC with a constant *z*_0_, the *η*_0_ of the blend must decrease by 32%. This did not differ greatly from the experimental values of PC-H/PMMA-L (90/10) and PMMA-H/PC-L (90/10).

### 3.2. Segregation Behavior during Injection Molding

Sample bars of PC-H/PMMA-L (90/10) and PMMA-H/PC-L (90/10) were prepared by injection molding. Both samples were transparent, as shown in Figure 7, suggesting that the miscibility did not change (phase separation did not take place).

The compositions of the blends at the surfaces of the samples were determined by obtaining their ATR-IR spectra and focusing on the intensities of the peaks attributable to carbonyl stretching vibrations, i.e., 1720–1730 cm^−1^ for PMMA and 1770–1780 cm^−1^ for PC. Prior to the measurements, the peak intensity ratios were determined using compression-molded films with various blend ratios to obtain a calibration curve. Because KRS-5 was used for the ATR crystal, the penetration depth (*d_P_*) of the IR beam, calculated using Equation (5) [44,45], was approximately 1 mm.
(5)$$d_{P}=\frac{\lambda /n_{1}}{2\pi \sqrt{\sin^{2}\theta -{\left(n_{2}/n_{1}\right)}^{2}}}$$
where *l* is the wavelength of the infrared beam, *n*_1_ and *n*_2_ are the refractive indices of the sample and the ATR crystal, respectively, and *q* is the incident angle of the IR.

Figure 8 compares the ATR-IR spectra of PMMA-H/PC-L (90/10) samples prepared by injection molding and compression molding. The vertical axis was normalized by the absorbance *A* of PMMA, i.e., *A*_PC_/*A*_PMMA_. The peak at 1770–1780 cm^−1^ for the injection-molded sample, which was obtained at point 3 in Figure 1 and can be ascribed to PC, is pronounced. The PC-L content of the injection-molded bar was estimated to be 12 wt.%, whereas that of the compression-molded film was 10 wt.%. These results suggest that surface segregation of PC-L, not phase separation, occurred during injection molding. Because of no phase separation (no interface to scatter the light), the product was transparent.

Figure 9 shows the ATR-IR spectra produced by the PC-H/PMMA-L (90/10) samples. The vertical axis was normalized to the absorbance *A* of PC, i.e., *A*_PMMA_/*A*_PC_. As in the previous figure, the peak intensity ascribed to PMMA-L differed between the injection-molded and compression-molded samples. The injection-molded bar had more PMMA-L on its surface (estimated to be 14 wt.%). This also suggests that surface segregation of the low molecular weight fraction occurred during injection molding.

Because the difference in the peak intensities was obvious in PC-H/PMMA-L (90/10), further ATR-IR measurements were performed at various positions using this sample. As shown in Figure 10, the PMMA-L content was high near the gate, suggesting that the segregation behavior was pronounced at high shear rates. At point 1, the PMMA-L content was calculated to be approximately 17 wt.%. Considering the transparency, the product did not show phase separation but had a concentration gradient as a function of the distance from the surface.

The detailed mechanism of the segregation behavior is unknown at present. However, at least, it did not originate from the adhesive nature with the mold surface, which must be determined by chemical structure, because a low molecular weight component, irrespective of the polymer species, has a large content on the surface. Once the low molecular weight fraction is segregated on the surface, i.e., the region with the highest shear rate, the shear stress would be largely reduced, leading to less hydrodynamic resistance for flow. This may be the origin of segregation, because the segregation was pronounced at high shear rates. The hypothesis can be checked by further experiments using other low molecular weight PMMA and/or PC samples having different molecular weight. On the other hand, the relationship with flow-induced demixing (phase separation), i.e., thermodynamic contribution, should be also considered. In this case, the degree of segregation must be reduced when the molecular weight of a low molecular weight component becomes low, because the mixing entropy is large. These experimental studies are currently being performed and will be reported soon.

From the viewpoint of industrial applications, an increase in the PMMA content on the surface is quite desirable because the surface hardness and scratch resistance will be greatly improved. Furthermore, the segregation of a low viscous component at the surface enhances the flowability, which is another desirable property of PC [15,46,47,48,49]. When the segregation is pronounced without light scattering, the product behaves like a multi-layered material such as plywood [50,51]. Such techniques will widen the material design of PC/PMMA blends and various other polymeric materials processed at high shear rates.

## 4. Conclusions

The flow induced segregation behavior of PC/PMMA blends was studied using samples with high and low molecular weights. Within the experimental range, the blends containing less than 20 wt.% PC-L and PMMA-L were miscible with PMMA-H and PC-H, respectively, at 250 °C. We investigated the surface compositions of the samples by ATR-IR spectroscopy using blend products prepared by injection molding and compression molding. The PC-L content was greater on the surfaces of the injection-molded bars made from the PMMA-H/PC-L (90/10) blends than on the surfaces of the compression-molded films made from the same blends. Furthermore, there was more PMMA-L on the surfaces of the injection-molded bars made from the PC-H/PMMA-L (90/10) blends than on the surfaces of the compression-molded films made from the same blends. Moreover, the segregation behavior of PMMA-L was most obvious near the gate, i.e., in the high shear rate region. These results suggest that the low molecular weight fraction of the miscible blends was segregated in the high shear rate region of the flow field. This will provide a novel technique for modifying surface properties.

## Figures and Tables

**Figure 1 materials-15-02994-f001:**
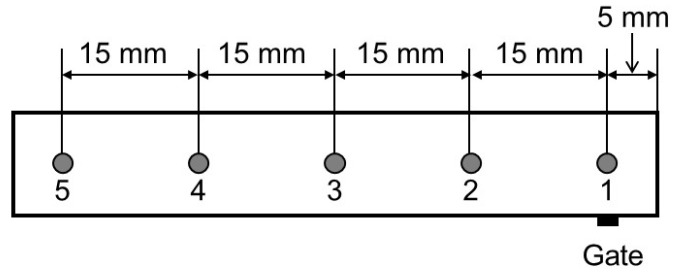
Measurement points used to obtain the attenuated total reflectance Fourier-transform infrared (ATR-IR) spectra of an injection-molded bar.

**Figure 2 materials-15-02994-f002:**
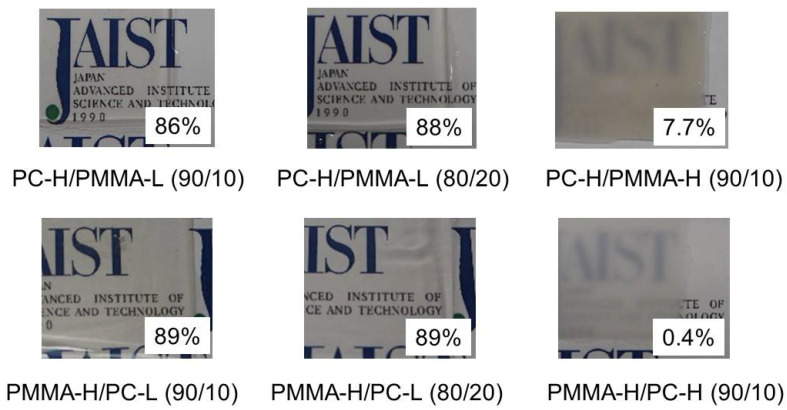
Photographs of sample films prepared by compression molding. The numbers in the figure represent the light transmittance at 589 nm.

**Figure 3 materials-15-02994-f003:**
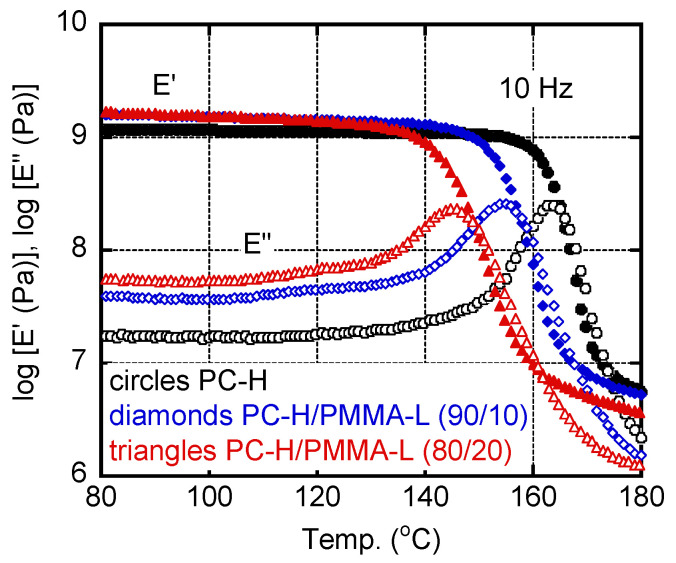
Temperature dependence of (closed symbols) tensile storage modulus *E′* and (open symbols) loss modulus *E″* at 10 Hz; (circles) PC-H, (diamonds) PC-H/PMMA-L (90/10), and (triangles) PC-H/PMMA-L (80/20).

**Figure 4 materials-15-02994-f004:**
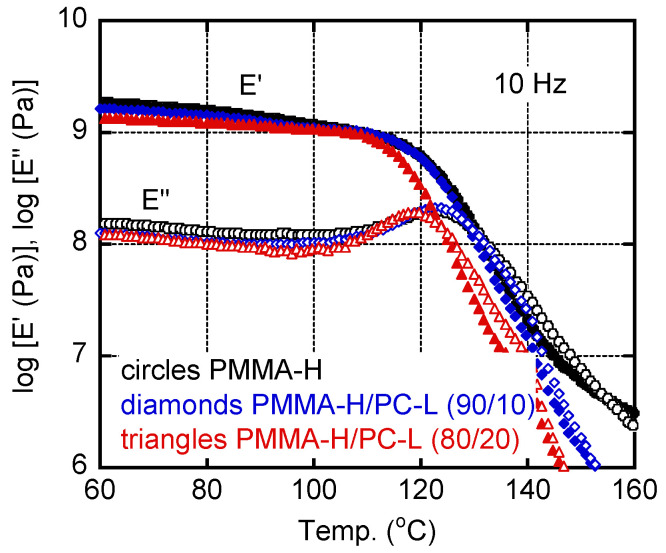
Temperature dependence of (closed symbols) tensile storage modulus *E′* and (open symbols) loss modulus *E″* at 10 Hz; (circles) PMMA-H, (diamonds) PMMA-H/PC-L (90/10), and (triangles) PMMA-H/PC-L (80/20).

**Figure 5 materials-15-02994-f005:**
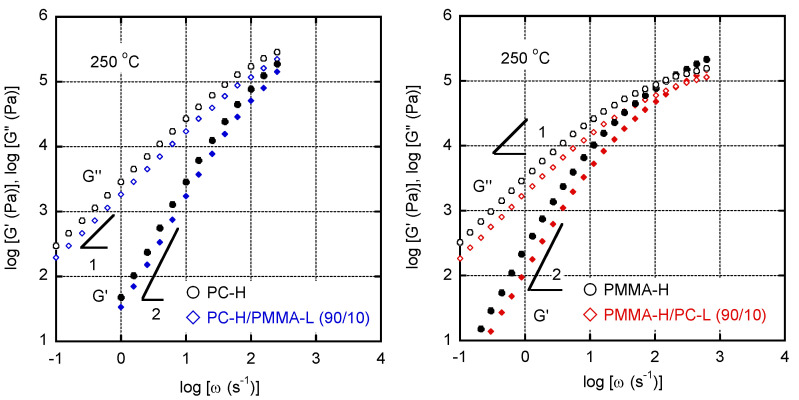
Angular frequency *w* dependence of the oscillatory shear moduli, i.e., the storage modulus *G′* and the loss modulus *G″*, at 250 °C of (**left**) PC-H and PC-H/PMMA-L (90/10) and (**right**) PMMA-H and PMMA-H/PC-L (90/10).

**Figure 6 materials-15-02994-f006:**
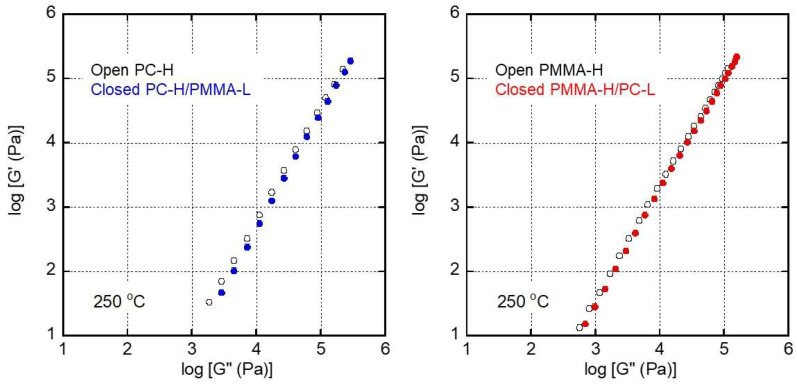
Han’s plot, i.e., the storage modulus *G′* plotted as a function of the loss modulus *G″*, at 250 °C of (**left**) PC-H and PC-H/PMMA-L (90/10) and (**right**) PMMA-H and PMMA-H/PC-L (90/10).

**Figure 7 materials-15-02994-f007:**
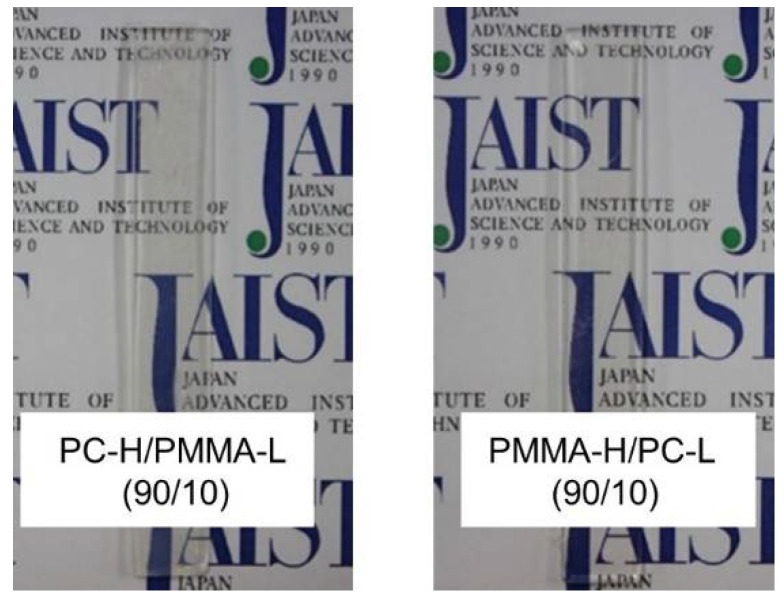
Injection-molded bars of PMMA-H/PC-L (90/10) and PC-H/PMMA-L (90/10).

**Figure 8 materials-15-02994-f008:**
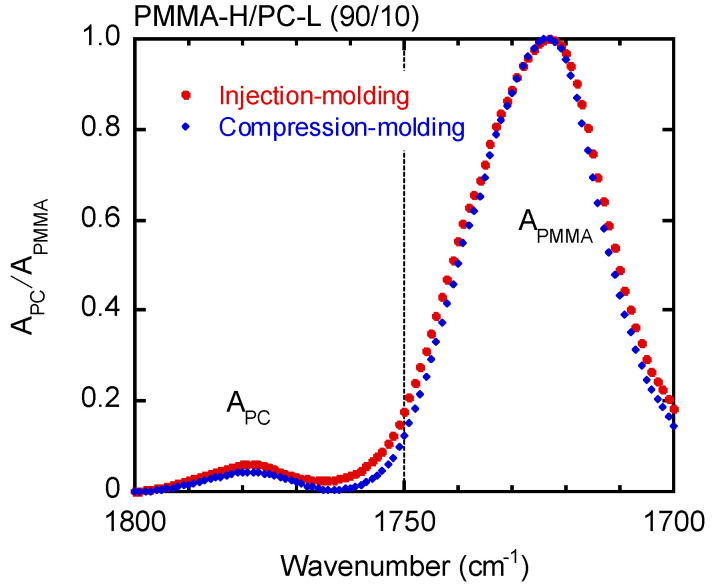
Normalized ATR-IR spectra of PMMA-H/PC-L (90/10) samples prepared by (circles) injection molding and (diamonds) compression molding.

**Figure 9 materials-15-02994-f009:**
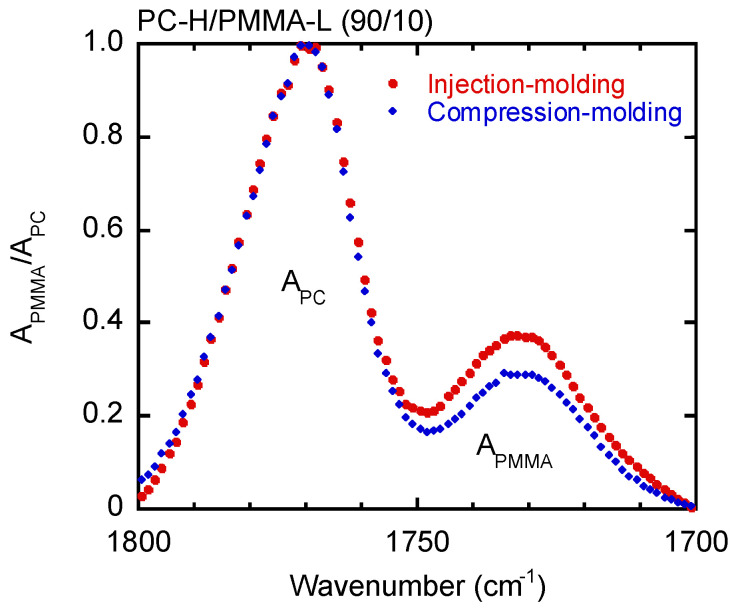
Normalized ATR-IR spectra of PC-H/PMMA-L (90/10) samples prepared by (circles) injection molding and (diamonds) compression molding.

**Figure 10 materials-15-02994-f010:**
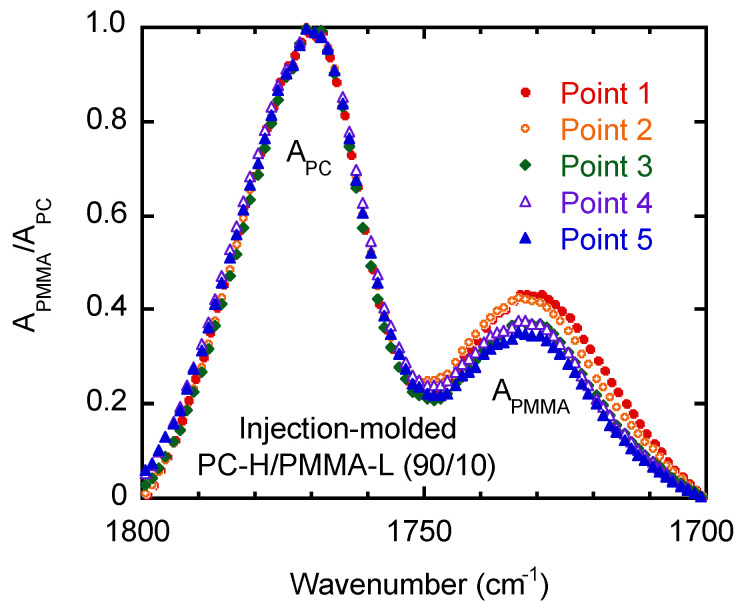
Normalized ATR-IR spectra at various positions of injection-molded PC-H/PMMA-L (90/10); (closed circles) point 1, (open circles) point 2, (diamonds) point 3, (open triangles) point 4, and (closed triangles) point 5.

**Table 1 materials-15-02994-t001:** Molecular weights of the polymers.

Sample Code	*M_n_*	*M_w_*	
PC-H	28,000	46,000	Panlite L-1225Y, Teijin, Japan
PC-L	3100	8700	Iupilon AL-071, Mitsubishi Engineering-Plastics, Japan
PMMA-H	58,000	120,000	Acrypet VH, Mitsubishi Chemical, Japan
PMMA-L	8900	15,000	Produced by Mitsubishi Chemical, Japan

## Data Availability

Not applicable.

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
