# Peer review of "Segregation Behavior of Miscible PC/PMMA Blends during Injection Molding"

_materials, 2022, doi:10.3390/ma15092994_

Round 1
Reviewer 1 Report
Reviewers comments:
This manuscript has been the focus of the surface segregation behaviour in miscible polymer blends (PC/PMMA). Miscible polymer blends were once considered a rarity. However, extensive research has led to the discovery of a large number of miscible polymer blends. Therefore, could you consider some points below for further improvement;
- Author might share or add some important data in the abstract i.e. The viscoelastic properties in the molten state.
- In partially miscible blends, a small part of one of the blend components is dissolved in the other part. This type of blend, which exhibits a fine phase morphology and satisfactory properties, is referred to as "compatible." Therefore, could the author provide the morphology of miscible polymer blends (PC/PMMA)?
Author Response
This manuscript has been the focus of the surface segregation behaviour in miscible polymer blends (PC/PMMA). Miscible polymer blends were once considered a rarity. However, extensive research has led to the discovery of a large number of miscible polymer blends. Therefore, could you consider some points below for further improvement;
[response]
Thank you for the reviewing process and positive comments.
- Author might share or add some important data in the abstract i.e. The viscoelastic properties in the molten state.
[response]
We added more explanations in the abstract including the viscoelastic properties.
- In partially miscible blends, a small part of one of the blend components is dissolved in the other part. This type of blend, which exhibits a fine phase morphology and satisfactory properties, is referred to as "compatible." Therefore, could the author provide the morphology of miscible polymer blends (PC/PMMA)?
[response]
Once there are two phases, they must be immiscible blends at which molar fraction of each component is predicted by lever rule (both polymers are partially dissolved each other even when phase separation occurs). In this study, all blends are miscible even for the blends having concentration gradation. To avoid misunderstanding, we revised some parts as follows.
[revised]
P.7, Line 232
Both samples were transparent, as shown in Figure 7, suggesting that the miscibility did not change (phase separation did not take place).
P.7, Line 253
These results suggest that surface segregation of PC-L, not phase separation (no interface to scatter the light), occurred during injection molding. Because of no phase separation, the product was transparent.
P.8, Line 276
Considering the transparency, the product did not show phase separation but had a concentration gradient as a function of the distance from the surface.
Reviewer 2 Report
The paper shows some interesting result about polymer blending process, but there are some major problems need to be solved before publication.
- The expression of the article is unclear, which should be improved wholly.
- The author represented that the aggregation of low-molecular-weight componenton the surface is responsible for the transparency of light, why? The explanation about mechanism is absent.
- Although ATR-IRtests could evaluate the content of groups in some extent, the slight gap is difficult to get a precise results. Are there other methods to prove it as complements? For example, the NMR?
- In many caes, the paper merely presents the phenomenon, but not the mechanism analyses, such as why the low-molecular-weight componentaggregates on the surface under flow field, is it ascribed to the special molecular structure?
- Some new references about the PC polymer blendingsuch as “Polymer-Plastics Technology and Materials, 2020, 59(9):998-1009.”could be cited.
Author Response
The paper shows some interesting result about polymer blending process, but there are some major problems need to be solved before publication.
1. The expression of the article is unclear, which should be improved wholly.
[response]
We tried to improve the whole content.
2. The author represented that the aggregation of low-molecular-weight component on the surface is responsible for the transparency of light, why? The explanation about mechanism is absent.
[response]
The refractive index difference is responsible for the light scattering. For the samples with surface segregation, however, there is not a clear interface to scatter the light.
[revised]
P.3, Line 118
Miscibility between PC and PMMA has been evaluated by light scattering [1-10]. There is a possibility to show good transparency of phase-separated blends when the component polymers show the same refractive indices [15,31]. However, the huge difference in the refractive index between PC and PMMA always resulted in light scattering when phase separation occurs. Therefore, it is highly difficult to obtain a transparent blend using only conventional polymers with high molecular weight.
P.7, Line 253
These results suggest that surface segregation of PC-L, not phase separation, occurred during injection molding. Because of no phase separation (no interface to scatter the light), the product was transparent.
3. Although ATR-IR tests could evaluate the content of groups in some extent, the slight gap is difficult to get a precise results. Are there other methods to prove it as complements? For example, the NMR?
[response]
At present, we cannot get any reliable results except for ATR-IR.
4. In many cases, the paper merely presents the phenomenon, but not the mechanism analyses, such as why the low-molecular-weight component aggregates on the surface under flow field, is it ascribed to the special molecular structure?
[response]
We added the followings in the revised version.
[revised]
P. 9, Line 286
The mechanism of the segregation behavior is unknown at present. It was not originated from the chemical structure because a low molecular weight component, irrespective of the polymer species, has a large content on surface. Once the low molecular weight fraction is segregated on the surface, i.e., the region with the highest shear rate, the shear stress would be largely reduced. This may be the origin of the segregation, because the segregation was pronounced at high shear rates. On the other hand, the relationship with flow-induced demixing should be also considered. These experimental studies are currently performed and will be reported soon.
5. Some new references about the PC polymer blending such as “Polymer-Plastics Technology and Materials, 2020, 59(9):998-1009.”could be cited.
[response]
We added it as ref. 49.
Reviewer 3 Report
This manuscript deals with preparation by injection molding and characterization of blends based on bisphenol-A polycarbonate (PC) and poly(methyl methacrylate) (PMMA). Mayor revision request.
Mayor revision:
- The morphology of the blends should be study by optical microscopy, scanning electron microscope, transmission electron microscope or atomic force microscopy. The transparency can be also related with the similar refractive index, this is why the morphology is request.
- How you prove the phase separation in this blends? The transparency is not enough to conclude phase separation?
3.Why authors do not perform DSC for confirm the Tg of the blends. Why you do not see both Tg, one correspond to PMMA and the other one for PC?
Minir revision
- Please extended the Abstract. Describe all your achievement in the Abstract to ensure interest of high number of Readers.
- The 20% of the reference is publications of authors. This should be justify.
Author Response
This manuscript deals with preparation by injection molding and characterization of blends based on bisphenol-A polycarbonate (PC) and poly(methyl methacrylate) (PMMA). Mayor revision request.
Mayor revision:
- The morphology of the blends should be study by optical microscopy, scanning electron microscope, transmission electron microscope or atomic force microscopy. The transparency can be also related with the similar refractive index, this is why the morphology is request.
[response]
As the reviewer commented, transparency shouldn’t be used to predict the miscibility when components have similar refractive indices. However, the blend system has a huge refractive index difference. Therefore, the optical transparency is good enough to predict the miscibility at least for PC/PMMA, as a number of researchers already revealed. Also we confirmed them by dynamic mechanical analysis and viscoelastic properties in the molten state. We believed this is good enough to prove the miscibility. Considering the comment from the reviewer, we added the explanation about them and introduced previous study on the PC/PMMA miscibility having similar molecular weights.
[revised]
P.3, Line 118
Miscibility between PC and PMMA has been evaluated by light scattering [1-10]. There is a possibility to show good transparency of phase-separated blends when the component polymers show the same refractive indices [15,31]. However, the huge difference in the refractive index between PC and PMMA always resulted in light scattering when phase separation occurs. Therefore, it is highly difficult to obtain a transparent blend using only conventional polymers with high molecular weight. Figure 2 shows the 500-µm-thick films prepared by compression molding. The films containing PC-L or PMMA-L were transparent, suggesting that they were miscible, at least at the compression-molding temperature, i.e., 250°C. According to our recent study using the PC/PMMA blends with similar molecular weights to the present PC-H/PMMA-L, the LCST of the blend containing 20 wt% of PMMA was 270°C [10].
P.4, Line 144
Miscibility between PC and PMMA has been also studied by dynamic mechanical analysis, because the glass-to-rubber transition is clearly detected for the system.
P.5, Line 193
Therefore, it can be concluded that the blend samples showed typical rheological behaviors as simple polymer melts without a long time relaxation mechanism ascribed to phase separation, demonstrating that they were miscible
- How you prove the phase separation in this blends? The transparency is not enough to conclude phase separation?
[response]
The response to this inquiry is the same with the previous one. Furthermore, we mentioned the difference between segregation and phase separation to avoid confusion.
[response]
P.7, Line 232
Both samples were transparent, as shown in Figure 7, suggesting that the miscibility did not change (phase separation did not take place).
P.7, Line 253
These results suggest that surface segregation of PC-L, not phase separation, occurred during injection molding. Because of no phase separation, the product was transparent.
P.8, Line 276
Considering the transparency, the product did not show phase separation but had a concentration gradient as a function of the distance from the surface.
- Why authors do not perform DSC for confirm the Tg of the blends. Why you do not see both Tg, one correspond to PMMA and the other one for PC?
[response]
It is well known that DSC is sensitive phase transformation but insensitive to glass transition. In contrast, dynamic mechanical properties provide the information on Tg quite clearly as demonstrated in the figure. Therefore, we strongly believe that DSC measurements are not required. The compression-molded samples were miscible, and thus there is a single peak in the E” curve. We added the following explanations.
[revised]
P.4, Line 144
Miscibility between PC and PMMA has been also studied by dynamic mechanical analysis, because the glass-to-rubber transition is clearly detected for the system.
P.4, Line 161
In the case of PC-H/PMMA-H (90/10), i.e., the opaque blend, double peaks were detected in the E” curve (but not presented here).
Minor revision
- Please extended the Abstract. Describe all your achievement in the Abstract to ensure interest of high number of Readers.
[response]
We extended the abstract following the suggestion.
- The 20% of the reference is publications of authors. This should be justify.
[response]
We removed some of our papers and added the papers from other research groups.
Reviewer 4 Report
The manuscript evaluates the blending between bisphenol-A polycarbonate (PC) and poly(methyl methacrylate) (PMMA) with variable molecular weights and evaluated their mechanical properties and surface segregation properties. The article is interesting and well written, however, it lacks information at several fronts. Therefore, I suggest authors to kindly consider the following suggestions and answer the concerns/questions for the greater clarity among readers:
- Please avoid the excessive self-citation. Currently, over 50% of the references belong to the corresponding author of this article. Kindly have a look and follow the journal policy for the same.
- In Figure 2, please check the transparency percentage for the figures. For instance, PC-H/PMMA-L appears less clear than PMMA-H/PCL-L.
- Please provide the reference for the Equation 3. It seems the Equation 3 needs a correction in terms of the refractive index term in the denominator. Please check/correct with appropriate reference.
- Kindly explain why the segregation of low viscous component in more pronounced at the surface. The current explanation on Page 7 line 234-235 probably needs further explanations.
- Did authors attempted to mix the equivalent molecular weight (preferably of low molecular weight) of PC and PMMA and evaluated their surface segregation behavior.
Author Response
The manuscript evaluates the blending between bisphenol-A polycarbonate (PC) and poly(methyl methacrylate) (PMMA) with variable molecular weights and evaluated their mechanical properties and surface segregation properties. The article is interesting and well written, however, it lacks information at several fronts. Therefore, I suggest authors to kindly consider the following suggestions and answer the concerns/questions for the greater clarity among readers:
- Please avoid the excessive self-citation. Currently, over 50% of the references belong to the corresponding author of this article. Kindly have a look and follow the journal policy for the same.
[response]
We removed some of our papers and added the papers from other research groups.
- In Figure 2, please check the transparency percentage for the figures. For instance, PC-H/PMMA-L appears less clear than PMMA-H/PCL-L.
[response]
Actually, they are almost the same. We cannot differentiate from one to the other. However, the surface reflectance is slightly different. We added the comment below.
[revised]
P.3, Line 131
The small difference between the PC-H blends and the PMMA-H blends must be attributed to the surface reflectance, which can be calculated by eq. (1) [32].
- Please provide the reference for the Equation 3. It seems the Equation 3 needs a correction in terms of the refractive index term in the denominator. Please check/correct with appropriate reference.
[response]
Thank you so much. This was our mistake. We revised the equation with references.
- Kindly explain why the segregation of low viscous component in more pronounced at the surface. The current explanation on Page 7 line 234-235 probably needs further explanations.
[response]
Thank you. We added our expectation, because we are not pretty sure the mechanism at present.
[revised]
P.9, Line 285
The mechanism of the segregation behavior is unknown at present. It was not originated from the chemical structure because a low molecular weight component, irrespective of the polymer species, has a large content on surface. Once the low molecular weight fraction is segregated on the surface, i.e., the region with the highest shear rate, the shear stress would be largely reduced. This may be the origin of the segregation, because the segregation was pronounced at high shear rates. On the other hand, the relationship with flow-induced demixing should be also considered. These experimental studies are currently performed and will be reported soon.
- Did authors attempted to mix the equivalent molecular weight (preferably of low molecular weight) of PC and PMMA and evaluated their surface segregation behavior.
[response]
We believe the viscosity is more important than the molecular weight. Since there is no meaning to compare the molecular weight (PS standard) of polymers with different chemical species. Therefore, we think we picked up appropriate samples in the present study.
Reviewer 5 Report
The manuscript by Moonprasith and coworkers mainly addresses the segregation behavior of injection-molded PC/PMMA blends. I suggest some major revisions before further consideration of the manuscript. Below are my points:
- The composition window prescribed for this paper is narrow. How about the cases with blends rich in PMMA-L and PC-L?
- Figure-2: names of the left columns seem wrong, please correct.
- The Tg for blends with varying compositions should be determined by DSC. Also, the experimental Tg should be compared with these predicted by Fox model, to show their miscibility.
- One cannot say the miscibility with the moduli dependent on frequency. Instead, Han plots or vGP curves should be further plotted to validate the miscibility from the rheological aspects.
- The profiles of viscosities of blends varying compositions should be presented.
- SEM images showing the surface segregation should be presented. As for this concern, one component at the surface should be selectively extracted by solvent before SEM observation. Meanwhile, micrographs showing the cross-sectional morphology should be also demonstrated.
- The concentration distribution along the thickness for low-Mw components should be demonstrated.
- The mechanism for the segregation behavior is still unclear. Please further clarify it with schemes.
- Mechanical properties for blends with varying compositions should be compared.
- How can we use this kind of segregation behavior to tailor the properties of such blends?
Author Response
The manuscript by Moonprasith and coworkers mainly addresses the segregation behavior of injection-molded PC/PMMA blends. I suggest some major revisions before further consideration of the manuscript. Below are my points:
- The composition window prescribed for this paper is narrow. How about the cases with blends rich in PMMA-L and PC-L?
[response]
Since PMMA-L and PC-L show very low viscosity, the blends containing a large amount of PMMA-L and PC-L cannot be used in industry. From the scientific point of view, it must be interesting. However, the blends must show very low viscosity and thus the experiments to apply homogeneous shear flow are difficult. Actually, we blended 50% of PC-L to PMMA-H, and found that they are immiscible.
- Figure-2: names of the left columns seem wrong, please correct.
[response]
Thank you so much. We revised them.
- The Tg for blends with varying compositions should be determined by DSC. Also, the experimental Tg should be compared with these predicted by Fox model, to show their miscibility.
[response]
It is well known that DSC is sensitive phase transformation but insensitive to glass transition. In contrast, dynamic mechanical properties provide the information on Tg quite clearly as demonstrated in the figure. Therefore, we strongly believe that DSC measurements are not required. The discussion on the Fox eq. was added as follows.
[revised]
P.4, Line 150
The peak temperature, i.e., the glass transition temperature Tg, decreased with increasing PMMA-L content; 163°C for PC-H, 155°C for PC-H/PMMA-L (90/10), and 145°C for PC-H/PMMA-L (80/20). Therefore, it can be concluded that the system is fully miscible, which agrees with previous studies using PC/PMMA blends in which molecular weight of one component was low [2,8-10]. It is well known that Tg of a miscible blend follows the Fox eq. [33], shown in eq. (2),
Eq. (2)
where Tg-i and wi are the Tg and weight fraction of i-th component, respectively.
Although Tg of PMMA-L was not detected by the measurement due to the brittle fracture of the film, eq. (2) predicted that it was 86°C. The Tg value was lower than that of a conventional PMMA [30,31], which is attributed to the low molecular weight as described in detail later. In the case of PC-H/PMMA-H (90/10), i.e., the opaque blend, double peaks were detected in the E” curve (but not presented here).
- One cannot say the miscibility with the moduli dependent on frequency. Instead, Han plots or vGP curves should be further plotted to validate the miscibility from the rheological aspects.
[response]
We added the Han’s plot in the revised version. Regarding the vGP, we did not use them since we measured the data at one temperature and further discussion using the same data should be avoided.
[revised]
P.5, Line 196
The corresponding Han’s plot [42], i.e., G’ plotted against G”, is shown in Figure 6. The straight lines were confirmed in the plot for both blends with a slightly low position as compared with the lines of the pure materials. These results also suggest that they were miscible at 250°C.
- The profiles of viscosities of blends varying compositions should be presented.
[response]
We measured the viscoelastic behavior of the samples in Figure 5. We denoted the values of zero-shear viscosity. Since the system is a simple polymer melt, we believe no need for the steady-state shear viscosity. The Cox-Merz should be applicable.
[revised]
P.6, Line 215
The values were 3000 (Pa s) for PC-H and 1950 (Pa s) for PC-H/PMMA-L (90/10). Therefore, the addition of 10 wt.% PMMA-L decreased the η0 by approximately 35%. The η0 of case of PMMA-H was 3100 (Pa s), whereas that of PMMA-H/PC-L (90/10) was 1950 (Pa s), i.e., a 37% decrease.
- SEM images showing the surface segregation should be presented. As for this concern, one component at the surface should be selectively extracted by solvent before SEM observation. Meanwhile, micrographs showing the cross-sectional morphology should be also demonstrated.
[response]
The system does not have phase separation. Once phase separation occurs, the system becomes opaque. No need for electron microscopic study. We added the following comments to insist them with previous reports.
[revised]
P.1, Line 36
In general, however, blends of PC and PMMA exhibit phase separation and consequent loss of transparency due the huge difference in the refractive indices of the components (PC 1.57 and PMMA 1.49) [1-7]. Although this makes easier to study the miscibility of PC/PMMA blends, the industrial application is greatly restricted.
P.3, Line 118
Miscibility between PC and PMMA has been evaluated by light scattering [1-10]. There is a possibility to show good transparency of phase-separated blends when the component polymers show the same refractive indices [15,31]. However, the huge difference in the refractive index between PC and PMMA always resulted in light scattering when phase separation occurs. Therefore, it is highly difficult to obtain a transparent blend using only conventional polymers with high molecular weight.
- The concentration distribution along the thickness for low-Mw components should be demonstrated.
[response]
Yes, we would like to know it. Although several measurements were performed using a microscopic FT-IR and ATR-IT with different ATR crystals, we have not obtained reliable data. This is our future plan.
- The mechanism for the segregation behavior is still unclear. Please further clarify it with schemes.
[response]
As the reviewer commented, the mechanism is still unclear. We added the following comments.
[revised]
P.9, Line 285
The mechanism of the segregation behavior is unknown at present. It was not originated from the chemical structure because a low molecular weight component, irrespective of the polymer species, has a large content on surface. Once the low molecular weight fraction is segregated on the surface, i.e., the region with the highest shear rate, the shear stress would be largely reduced. This may be the origin of the segregation, because the segregation was pronounced at high shear rates. On the other hand, the relationship with flow-induced demixing should be also considered. These experimental studies are currently performed and will be reported soon.
- Mechanical properties for blends with varying compositions should be compared.
[response]
We did not do the mechanical tests. This is the miscible system, therefore, we cannot expect a huge change in the mechanical properties. At least in this paper, we should focus on the segregation behavior. When the mechanical properties are listed, the purpose must be obscure, and the value of this paper becomes low.
- How can we use this kind of segregation behavior to tailor the properties of such blends?
[response]
We added something about applications.
[revised]
P.9, Line 293
From the viewpoint of industrial applications, an increase in the PMMA content on the surface is quite desirable because the surface hardness and scratch resistance will be greatly improved. Furthermore, the segregation of a low viscous component at the surface enhances the flowability, which is another desirable property of PC [15,46-48]. When the segregation is pronounced without light scattering, the product behaves like a multi-layered material such as plywood [49,50]. Such techniques will widen the material design of PC/PMMA blends and various other polymeric materials processed at high shear rates.
Round 2
Reviewer 2 Report
The paper could be accepted in present form.
Author Response
The paper could be accepted in present form.
[response]
Thank you very much.
Reviewer 4 Report
Author's have revised the manuscript moderately but lacks the mechanistic details for the observations.
Furthermore, in answer to one of the reviewer's question, which is as follows:
- Did authors attempted to mix the equivalent molecular weight (preferably of low molecular weight) of PC and PMMA and evaluated their surface segregation behavior.
Author's responded that viscosity is more important than the molecular weight and the samples that have been used in this study are appropriate.
First of all, the molecular weight and viscosity are directly related and therefore the present question may not be disregarded with the given explanation in the response sheet.
Secondly, the raised concern does not question the current sample appropriateness but rather seeks an explanation regarding which polymer type may get competitive advantage for surface segregation if both of the polymers are of low and similar viscosity with appreciable optical clarity.
Author Response
Author's have revised the manuscript moderately but lacks the mechanistic details for the observations. Furthermore, in answer to one of the reviewer's question, which is as follows:
5. Did authors attempted to mix the equivalent molecular weight (preferably of low molecular weight) of PC and PMMA and evaluated their surface segregation behavior.
Author's responded that viscosity is more important than the molecular weight and the samples that have been used in this study are appropriate.
First of all, the molecular weight and viscosity are directly related and therefore the present question may not be disregarded with the given explanation in the response sheet.
Secondly, the raised concern does not question the current sample appropriateness but rather seeks an explanation regarding which polymer type may get competitive advantage for surface segregation if both of the polymers are of low and similar viscosity with appreciable optical clarity.
[response]
Thank you very much for considerable comments. As the reviewer said, the viscosity is proportional to Mw (Mw<Mc) or Mw^3.4 (Mw>Mc) for the same polymer species. However, this relationship is not applicable when the polymer species is different. Besides, the molecular weights evaluated by GPC were relative ones as compared with the standard PS samples. Since the spatial size of molecules in a specific solvent, i.e., chloroform, depends on the polymer species, the direct comparison of molecular weights between PC and PMMA has less meaning. This is why we answered that the viscosity is more important than the molecular weight (but lack of the explanation, sorry). Strictly speaking, we should evaluate the absolute value of molecular weight for all samples. Then we can discuss the contribution of mixing entropy.
We also understand the comment from the reviewer about the explanation on “which polymer type may get competitive advantage for surface segregation…”. This is a very important point. We appreciate the comment.
Because we cannot clarify the mechanism yet, we added the following explanation in the revised version.
[revised]
P.9, Line 285
The detailed mechanism of the segregation behavior is unknown at present. However, at least, it was not originated from the adhesive nature with the mold surface, which must be determined by chemical structure, because a low-molecular-weight component, irrespective of the polymer species, has a large content on surface. Once the low-molecular-weight fraction is segregated on the surface, i.e., the region with the highest shear rate, the shear stress would be largely reduced, leading to less hydrodynamic resistance for flow. This may be the origin of segregation, because the segregation was pronounced at high shear rates. The hypothesis can be checked by further experiments using other low-molecular-weight PMMA and/or PC samples having different molecular weight. On the other hand, the relationship with flow-induced demixing (phase separation), i.e., thermodynamic contribution, should be also considered. In this case, the degree of segregation must be reduced when the molecular weight of a low-molecular-weight component becomes low, because the mixing entropy is large. These experimental studies are currently performed and will be reported soon.
Reviewer 5 Report
With my questions answered by the authors, the article can be accepted.
Author Response
With my questions answered by the authors, the article can be accepted.
[response]
Thank you very much.